# KALFormer: Knowledge-augmented attention learning for long-term time series forecasting with transformer

Xing Dong[1], Qianwei Yang[2], Wenbo Cheng[3], Yun Zhang[3]*

1 School of Basic Medical Sciences, Guizhou University of Traditional Chinese Medicine, Guiyang, China, 2 National Engineering Laboratory for Big Data System Computing Technology, Shenzhen University, Shenzhen, China, 3 School of Information Engineering, Guizhou University of Traditional Chinese Medicine, Guiyang, China

* zhangyun016@gzy.edu.cn

## Abstract

Time series forecasting remains a fundamental yet challenging task due to its inherent non-linear dynamics, inter-variable dependencies, and long-term temporal correlations. Existing approaches often struggle to jointly capture local temporal continuity and global contextual relationships, particularly under complex external influences. To overcome these limitations, we propose KALFormer, a knowledge-augmented attention learning transformer framework that integrates sequential modeling with external information fusion. KALFormer enhances spatiotemporal representation and contextual reasoning by integrating Long Short-Term Memory (LSTM) encoders, Transformer-based self-attention mechanisms, and knowledge-aware modules. Extensive experiments on six public benchmark datasets demonstrate that KALFormer achieves an average improvement of 8.4% in MSE and MAE compared with representative baseline models, highlighting its robustness, interpretability, and reliability for long-term time series forecasting. The source code is available at https://github.com/dxpython/KALFormer.

## 1 Introduction

Time series forecasting is a critical task in a wide range of fields [1], where accurate prediction of future trends is essential. The complexity of time series data [2] and the challenge of capturing long-range dependencies [3] are fundamental difficulties that have persisted in this domain. As the length of sequences grows, the contextual relationships between distant time steps become increasingly intricate. Traditional models, such as Long Short-Term Memory (LSTM) [22–24] and Recurrent Neural Networks (RNN) [4], often exhibit limitations in capturing these distant dependencies, leading to reduced forecasting accuracy, particularly when dealing with long-range patterns.

**Data availability statement:** Data download: https://doi.org/10.5281/zenodo.17068599.

**Funding:** This work was financially supported by the Undergraduate Teaching Reform Project (2024-26) from Guizhou University of Traditional Chinese Medicine, the Doctoral Start-up Fund (2019-76) from Guizhou University of Traditional Chinese Medicine, and the National Natural Science Foundation of China (Grant No. 32360152).

**Competing interests:** The authors have declared that no competing interests exist.

Additionally, in many real-world applications, forecasting tasks involve multiple correlated variables that evolve under the influence of external and contextual factors [5]. For example, policy changes, economic trends, and environmental conditions such as weather jointly affect multiple variables in domains like finance [7] and energy management [6]. Therefore, it is crucial to model these inter-variable dependencies and nonlinear interactions rather than treating them as isolated exogenous influences. This motivates a multivariate forecasting formulation, where all variables are predicted jointly to capture both temporal dynamics and cross-variable relationships [32].

To tackle these issues, we propose KALFormer, Knowledge-Augmented Attention Learning for Long-Term Time Series Forecasting with Transformer. This model integrates LSTM, Knowledge Augmented Network (KAN) [29], Multi-Head Attention (MHA) [12], and Transformer architecture [26], addressing both long-range dependencies and disruptive factors. Specifically, the KAN component enhances the model's ability to integrate domain-informed external factors, while the Transformer architecture—with its multi-head attention mechanism and feedforward networks— facilitates the efficient fusion of features across different time steps.

In summary, our contributions are as follows:

- A unified framework, KALFormer, is developed by integrating LSTM, self-attention, KAN, and Transformer modules to address contextual dependencies and external interferences in long-term time series forecasting.
- The incorporation of knowledge-augmented representations through KAN enhances the model's interpretability and environmental awareness, enabling more reliable prediction in complex domains such as finance and energy.
- Comprehensive experiments on multiple benchmark datasets demonstrate that KALFormer achieves consistently improved predictive accuracy and robustness compared with existing forecasting approaches.

## 2 Related work

### 2.1 Long-term dependency models for series prediction

**2.1.1 RNN-based and CNN-based methods.** RNNs [4] and their variants, such as LSTM [22–24], more comprehensive reviews of RNN architectures are provided in [25,30] and Gated Recurrent Units (GRU) [8], are among the earliest deep learning architectures applied to time series forecasting. Due to their recursive structure, RNNs process sequential data step by step, updating hidden states at each time step to capture temporal dependencies [9–11]. This characteristic makes them well-suited for modeling short- to medium-term dependencies in time series data.

Several improvements have been proposed to enhance the performance of RNN-based models. For instance, Sagheer et al. introduced a deep Long Short-Term Memory (DLSTM) model [9], where a genetic algorithm was employed to optimize the model architecture. When evaluated on oil industry data, the DLSTM model outperformed various statistical and computational intelligence techniques, demonstrating its effectiveness in capturing both past and future dependencies [31].

Similarly, Chang et al. developed the Memory Time-series Network (MTNet) [10], which integrates a large memory module, three independent encoders, and an autoregressive component, thereby improving the model's ability to learn from long-term historical data.

Despite these advancements, RNN-based methods face inherent challenges, particularly when dealing with very long sequences. One major limitation is their difficulty in effectively capturing long-range dependencies due to information dissipation during backpropagation. Although LSTM and GRU partially address this issue through gating mechanisms, they still struggle when processing extremely long sequences [16]. Additionally, the sequential nature of RNNs restricts their ability to leverage parallel computation, resulting in inefficient training and slower inference times for large-scale time series data.

To mitigate these issues, alternative architectures, such as convolutional neural networks (CNNs), have been explored for time series forecasting. CNNs can efficiently extract local temporal patterns using convolutional filters, enabling faster computations and improved scalability compared to RNNs. However, CNNs alone often lack the ability to capture long-term dependencies without additional modifications, such as dilated convolutions or hybrid models that integrate recurrent structures.

More recently, TimesNet [36] has emerged as an alternative approach that extends CNN-based methods by focusing on temporal 2D-variation modeling. Unlike conventional sequence-based models, TimesNet represents time series as 2D tensors, allowing it to effectively analyze temporal data with non-uniform frequencies and resolutions. This makes it particularly advantageous in scenarios where traditional RNNs and CNNs struggle with irregular time series patterns. By leveraging its novel representation, TimesNet outperforms traditional models in handling dynamic temporal patterns and provides a scalable solution for forecasting applications.

In summary, while RNNs and their variants have played a crucial role in time series forecasting, their inherent limitations in handling long-term dependencies and computational efficiency have motivated the development of alternative approaches, including CNN-based models, TimesNet, attention-based models, and hybrid architectures.

**2.1.2 Attention-based methods.** To overcome the inherent limitations of RNN architectures, the Attention mechanism was introduced, emerging as a powerful tool for time series modeling, particularly in solving long-range dependency issues. By calculating the relevance between different time steps within an input sequence, Attention enables models to identify which time points are crucial for the current prediction. This allows models to focus on the most relevant information without being constrained by the step-by-step processing inherent in RNNs, thereby mitigating the vanishing gradient problem and inefficiencies associated with sequential data processing.

One of the prominent Attention-based models is the Transformer [26], which fully leverages the Attention mechanism and completely eliminates the need for recurrent structures. By employing parallel computations, the Transformer architecture [27] dramatically enhances training efficiency. Its Multi-Head Attention mechanism [12] allows the model to attend to different subspaces of the input, thus capturing a wide array of global dependencies and providing a more comprehensive understanding of long-range temporal relationships.

Although the Transformer architecture has shown strong capability in modeling long-range dependencies, it still faces limitations in capturing complex spatiotemporal interactions and non-stationary dynamics that often occur in real-world time series. To address these challenges, several Transformer-based variants have been proposed to enhance representation of temporal and cross-variable relationships. For example, the iTransformer [33] adopts an inverted architecture to improve modeling of long-range dependencies through self-attention mechanisms, while the Crossformer [34] introduces a cross-dimension dependency module that jointly captures temporal and spatial correlations in multivariate series. Likewise, the Deep Time-Index model [35] learns time-indexed representations to balance short- and long-term dependencies, thereby improving forecasting performance across diverse domains.

Informer [14] further refines Transformer-based time series forecasting by addressing the high complexity and memory usage issues associated with long-sequence processing. It introduces the ProbSparse self-attention mechanism, self-attention distillation, and a generative decoder, which significantly improve computational efficiency while maintaining

strong predictive capabilities. Non-stationary Transformers [15] further advance this paradigm by incorporating both stationary and non-stationary attention mechanisms, allowing the model to adaptively balance predictability and flexibility, which is crucial for forecasting non-stationary data.

The VMD-Crossformer [13] is proposed for short-term power load forecasting. By combining Variational Mode Decomposition (VMD) with the Crossformer architecture, the model effectively captures both temporal and dimensional dependencies. The Two-Stage Attention layer in Crossformer is designed to simultaneously capture dependencies across time and feature dimensions, improving forecasting performance.

Collectively, these Attention-based methods represent significant advancements in time series forecasting, each employing unique strategies to handle complex, high-dimensional temporal data. By leveraging self-attention mechanisms, they offer improved efficiency, scalability, and forecasting accuracy over traditional RNN-based approaches, making them highly effective for a wide range of real-world applications.

### 2.2 Handling external factors in time series prediction

In practical applications, the dynamic behavior of time series data is often shaped not only by historical trends but also by various other factors. For example, fluctuations in electricity load may be affected by weather conditions, social activities, and holidays. Consequently, relying solely on historical data for time series modeling can be insufficient for capturing the complex interplay between these influencing elements.

To address these complexities more effectively, KAN has been applied to time series forecasting [29]. KAN enhances the model's interpretive ability by integrating external knowledge bases or graph data into the forecasting process. These additional sources may include domain-specific expertise, industry standards, or other relevant contextual information. By leveraging Graph Convolutional Networks (GCN) [19–21], this information is encoded into low-dimensional embeddings that are subsequently fused with historical time series data.

In this context, GCNs model the relationships between influencing factors, generating contextually meaningful embedding vectors. These embeddings, when combined with time series features, enable the model to leverage both historical data and a deeper understanding of the broader influences shaping the predictions. For example, Han et al. [17] introduced the Graph Hawkes Neural Network, designed to capture complex dependencies within dynamic sequences and predict future events. Experimental results on large-scale temporal multi-relational datasets validated its effectiveness. Similarly, Zhong et al. [18] proposed the Knowledge Graph-Augmented Network for fine-grained sentiment analysis, showcasing how contextual knowledge can be efficiently combined with linguistic data. Their method integrated multiple perspectives, including syntax, context, and domain information, to extract sentiment features with greater accuracy.

While GCNs excel at modeling the underlying relationships between influencing factors, they aggregate all neighboring nodes with uniform weights, which limits their capacity to model heterogeneous or asymmetric dependencies among variables [28]. To overcome this limitation, our framework replaces static GCN aggregation with a graph attention mechanism, allowing the model to dynamically learn the relative importance of neighboring nodes. This adaptive weighting enables more flexible and context-aware information propagation, thereby improving the expressiveness and robustness of the knowledge integration process.

## 3 Method

### 3.1 Overall architecture

In time series forecasting, models typically face three fundamental challenges. First, sequence data are inherently complex, with multidimensional features and nonlinear relationships that hinder traditional approaches from capturing intricate dynamic patterns. Second, time series often exhibit long-term dependencies, and models frequently struggle to preserve

distant historical information, which reduces prediction accuracy for long sequences. Third, external influences such as economic trends and environmental changes further complicate forecasting, increasing task difficulty.

To address these issues, we propose the KALFormer model, Knowledge-Augmented Attention Learning for Long-Term Time Series Forecasting with Transformer. KALFormer integrates LSTM [22], Transformer [26], MHA, and the KAN [29] to capture long-range dependencies and complex temporal patterns more effectively. The overall architecture is illustrated in Fig 1.

## 3.2 Sequential backbone with LSTM and attention

Recurrent architectures are widely adopted for sequential modeling due to their ability to preserve contextual information across time. Among them, LSTM mitigate the vanishing gradient problem by introducing gating mechanisms that regulate information flow. Nevertheless, standard LSTMs often struggle to capture very long-range dependencies and complex temporal dynamics, which limits their capacity for modeling global context. To address this issue, we employ a sequential backbone that combines stacked LSTM layers with a self-attention mechanism, thereby enabling the extraction of both local continuity and long-distance dependencies.

Formally, let $X \in \mathbb{R}^{N \times D}$ denote the input sequence, where $N$ is the number of time steps and $D$ is the feature dimension. Temporal features are obtained by passing $X$ through two stacked LSTM layers:

$$z = \text{LSTM}_2(\text{LSTM}_1(X)). \tag{1}$$

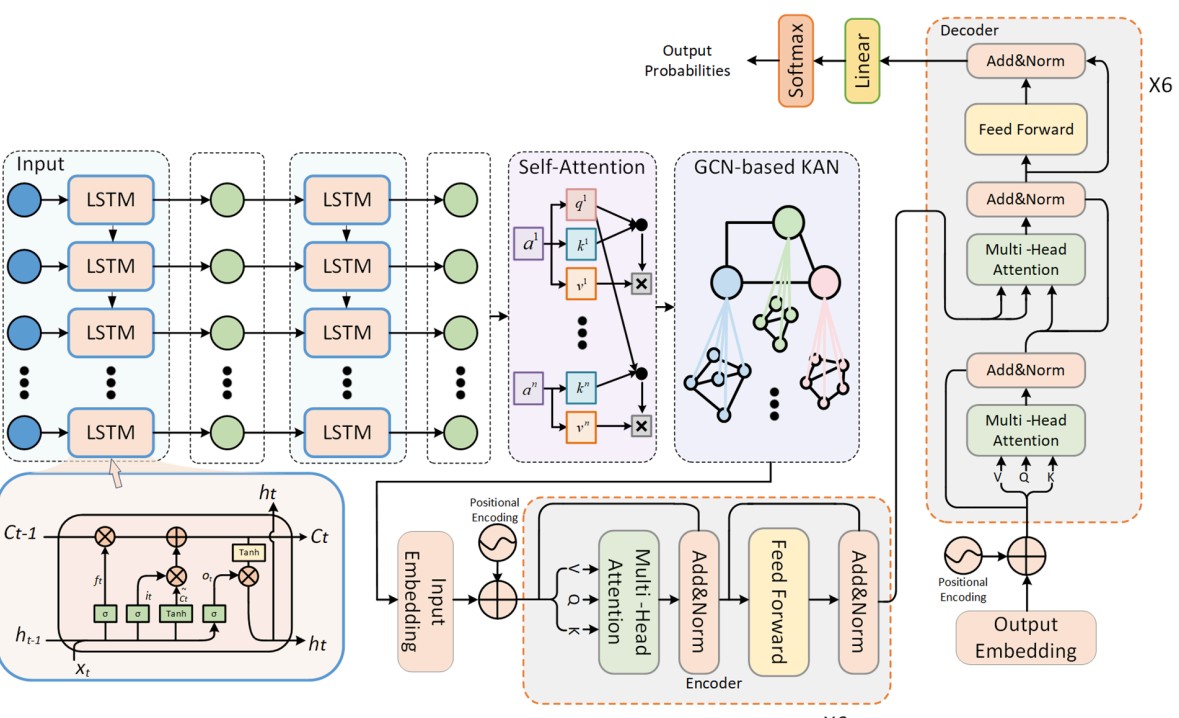

**Fig 1. Overall architecture of KALFormer.** The framework integrates Long Short-Term Memory (LSTM) units for temporal encoding, a self-attention mechanism for capturing global dependencies, a Graph Neural Network (GNN)-based Knowledge-Augmented Network (KAN) for nonlinear relational interactions, and a multi-layer Transformer encoder–decoder for feature fusion and sequence prediction. The model outputs are normalized and passed through a linear layer and Softmax for forecasting probabilities.

The memory update within each LSTM cell is expressed compactly as

$$\mathbf{c}_t = \mathbf{f}_t \odot \mathbf{c}_{t-1} + \mathbf{i}_t \odot \tilde{\mathbf{c}}_t, \qquad \mathbf{h}_t = \mathbf{o}_t \odot \tanh(\mathbf{c}_t), \tag{2}$$

where $\mathbf{i}_t, \mathbf{f}_t, \mathbf{o}_t \in (0,1)^d$ denote the input, forget, and output gates, respectively, and $\tilde{\mathbf{c}}_t \in \mathbb{R}^d$ is the candidate memory cell. The vector $\mathbf{c}_t$ represents the internal cell state that carries long-term information, while $\mathbf{h}_t \in \mathbb{R}^d$ denotes the hidden state at time step $t$, which serves as both the output of the current cell and the input to subsequent layers or future time steps. This formulation highlights the gated mechanism that allows the LSTM to preserve long-term dependencies while adaptively integrating new temporal information.

To complement sequential modeling, a self-attention layer is applied to the hidden representations $\mathbf{H} = \{\mathbf{h}_1, \ldots, \mathbf{h}_N\}$. In this mechanism, each time step interacts with all others through a set of learned projections that generate a query–key–value (QKV) triplet. Specifically, the hidden sequence is linearly mapped to query, key, and value matrices as $\mathbf{Q} = \mathbf{H}W_Q$, $\mathbf{K} = \mathbf{H}W_K$, and $\mathbf{V} = \mathbf{H}W_V$, where $W_Q$, $W_K$, and $W_V$ are trainable weight matrices. The attention-enhanced representation is then computed as

$$\text{Attention}(\mathbf{Q}, \mathbf{K}, \mathbf{V}) = \text{softmax}\left(\frac{\mathbf{Q}\mathbf{K}^\top}{\sqrt{d_k}}\right)\mathbf{V}, \tag{3}$$

where $d_k$ denotes the key dimension. This mechanism assigns adaptive weights to temporal positions, enabling the model to highlight informative time steps while suppressing redundant or less relevant ones.

By integrating LSTM and self-attention, the backbone leverages the complementary strengths of both modules: the LSTM ensures sequential continuity and memory preservation, whereas the attention mechanism dynamically highlights salient dependencies irrespective of temporal distance. The resulting representation serves as a robust foundation for subsequent knowledge integration and fusion.

### 3.3 Knowledge integration through knowledge-augmented network

Although recurrent and attention-based encoders effectively capture temporal dependencies, they remain limited in explicitly modeling the cross-variable interactions that characterize multivariate time series. To address this limitation, a KAN is introduced to integrate structured relational information into the representation learning process.

Specifically, a variable graph is defined as $G = (\mathcal{V}, \mathcal{E}, \mathbf{A})$, where $\mathcal{V}$ denotes the set of variables (nodes), $\mathcal{E}$ represents the set of edges encoding statistical or semantic relations among variables, and $\mathbf{A}$ is the weighted adjacency matrix that quantifies connection strengths. Each node $v_j \in \mathcal{V}$ corresponds to one feature dimension in the multivariate input sequence $X \in \mathbb{R}^{N \times D}$, where $N$ is the sequence length (time steps) and $D$ is the feature dimension, i.e., the number of observed variables or sensors in the dataset. The value of $D$ is determined directly by the dataset configuration (e.g., the number of sensors in traffic data or meteorological indicators in weather data) and remains fixed during both training and inference, ensuring a consistent graph structure for the KAN across all temporal windows.

The adjacency matrix combines statistical correlation with domain priors in a convex form,

$$\mathbf{W} = \eta\left(\lambda\,\mathbf{S}_+ + (1-\lambda)\,\mathbf{P}\right), \qquad \lambda = \sigma(\gamma) \in (0,1), \tag{4}$$

where $\mathbf{S} \in \mathbb{R}^{D \times D}$ is the Pearson correlation matrix estimated from training data, $\mathbf{S}_+ = \max(\mathbf{S}, 0)$ ensures nonnegativity, $\mathbf{P}$ encodes domain-specific structural priors (e.g., sensor connectivity in traffic or meteorological coupling patterns), $\eta(\cdot)$ denotes row-wise normalization, and $\lambda$ is a learnable mixing coefficient. To encourage sparsity, only the top-$k$ strongest connections per node are retained,

$$\mathbf{A} = \text{TopK}(\mathbf{W}, k), \tag{5}$$

followed by the addition of self-loops and symmetric normalization,

$$\tilde{\mathbf{A}} = \mathbf{A} + \mathbf{I}, \qquad \hat{\mathbf{A}} = \mathbf{D}^{-1/2}\tilde{\mathbf{A}}\mathbf{D}^{-1/2}, \quad \mathbf{D} = \text{diag}(\tilde{\mathbf{A}}\mathbf{1}), \tag{6}$$

which yields a normalized adjacency matrix $\hat{\mathbf{A}}$ for stable propagation.

Each node $v_j$ is initialized with an embedding $\mathbf{e}_j$, collected in $\mathbf{E} \in \mathbb{R}^{D \times d_0}$. Information is propagated through a two-layer graph neural network (GNN) based on neighborhood aggregation,

$$\mathbf{G}^{(1)} = \sigma(\hat{\mathbf{A}}\mathbf{E}\mathbf{W}^{(0)}), \qquad \mathbf{K} = \sigma(\hat{\mathbf{A}}\mathbf{G}^{(1)}\mathbf{W}^{(1)}), \tag{7}$$

where $\{\mathbf{W}^{(0)}, \mathbf{W}^{(1)}\}$ are trainable weights and $\sigma(\cdot)$ denotes a nonlinearity. The output $\mathbf{K} \in \mathbb{R}^{D \times d_g}$ serves as knowledge embeddings that encode cross-variable dependencies and domain-informed relationships.

To condition temporal features on these knowledge embeddings, cross-attention is applied with queries from the temporal representation $\mathbf{H} \in \mathbb{R}^{N \times d_h}$ and keys/values from $\mathbf{K}$,

$$\mathbf{Q} = \mathbf{H}\mathbf{W}_Q, \quad \mathbf{K}_K = \mathbf{K}\mathbf{W}_K, \quad \mathbf{K}_V = \mathbf{K}\mathbf{W}_V, \qquad \mathbf{C} = \text{softmax}\left(\frac{\mathbf{Q}\mathbf{K}_K^{\top}}{\sqrt{d_k}}\right)\mathbf{K}_V, \tag{8}$$

producing a knowledge-informed context $\mathbf{C} \in \mathbb{R}^{N \times d_v}$. The final integration employs a learnable gating mechanism,

$$\boldsymbol{\alpha} = \sigma(\mathbf{W}_\alpha[\bar{\mathbf{H}}; \bar{\mathbf{K}}] + \mathbf{b}_\alpha) \in (0, 1)^{d_v}, \qquad \tilde{\mathbf{H}} = \mathbf{H} + \boldsymbol{\alpha} \odot \mathbf{C}, \tag{9}$$

where $\bar{\mathbf{H}} = \frac{1}{N}\sum_{t=1}^{N}\mathbf{h}_t$ and $\bar{\mathbf{K}} = \frac{1}{D}\sum_{j=1}^{D}\mathbf{k}_j$ denote global summaries. The gate $\boldsymbol{\alpha}$ is initialized to 0.5 and optimized jointly with the rest of the model, balancing contributions from sequential and knowledge pathways.

### 3.4 Representation fusion with multi-head transformer

Although knowledge-conditioned features enhance cross-variable modeling, a single attention mechanism may still be insufficient to capture the diverse dependency patterns present in long and heterogeneous sequences. To enrich representational capacity, we incorporate a MHA mechanism, which projects the input into multiple subspaces and performs attention operations in parallel. This design allows the model to simultaneously attend to distinct aspects of temporal and relational structure, thereby improving robustness and generalization. The overall architecture is illustrated in Fig 2.

Formally, given the hidden representation $\tilde{\mathbf{H}} \in \mathbb{R}^{N \times d_h}$ after knowledge integration, the query, key, and value matrices for the $m$-th head are defined as

$$\mathbf{Q}_m = \tilde{\mathbf{H}}W_Q^{(m)}, \quad \mathbf{K}_m = \tilde{\mathbf{H}}W_K^{(m)}, \quad \mathbf{V}_m = \tilde{\mathbf{H}}W_V^{(m)}, \tag{10}$$

where $W_Q^{(m)}, W_K^{(m)}, W_V^{(m)} \in \mathbb{R}^{d_h \times d_k}$ are head-specific projection matrices. The output of each head is computed using scaled dot-product attention:

$$\text{head}_m = \text{softmax}\left(\frac{\mathbf{Q}_m\mathbf{K}_m^{\top}}{\sqrt{d_k}}\right)\mathbf{V}_m. \tag{11}$$

Outputs from all $M$ heads are concatenated and linearly transformed:

$$\text{MHA}(\tilde{\mathbf{H}}) = \text{Concat}(\text{head}_1, \dots, \text{head}_M)W_O, \tag{12}$$

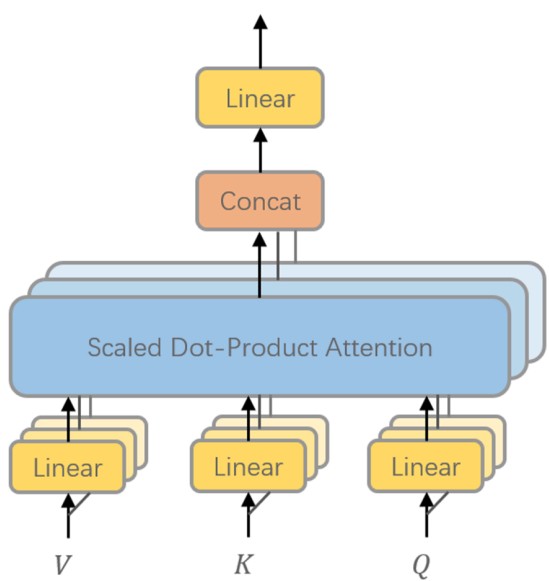

**Fig 2**. **Structure of the multi-head attention mechanism.** Each attention head performs a scaled dot-product attention using individual query (*Q*), key (*K*), and value (*V*) projections. The outputs from all heads are concatenated and linearly transformed to form the final attention representation, enabling the model to jointly attend to information from different subspaces.

with $W_O \in \mathbb{R}^{Md_v \times d_h}$ denoting the output projection. This formulation enables the network to capture complementary patterns from multiple representation subspaces in parallel.

By integrating multi-head attention, the model leverages diverse relational cues and contextual information across temporal positions, providing a more expressive fusion of sequential and knowledge-informed representations. This enriched feature space serves as the foundation for subsequent prediction layers.

## 4 Experiment

### 4.1 Data foundations and evaluation methodology

We evaluated the proposed model on six real-world benchmark datasets—Traffic, Weather, Electricity, ETTh1, ETTh2, ETTm1, and ETTm2 as summarized in Table 1. Their temporal resolutions are hourly for Traffic, Electricity, ETTh1, and ETTh2, 10 minutes for Weather, and 5 minutes for ETTm1 and ETTm2. Each dataset was reformulated as a graph, where nodes represent variables and edges encode statistical associations or domain-specific priors, enabling expressive spatiotemporal learning. All datasets are publicly available at Zenodo (https://doi.org/10.5281/zenodo.17068599).

**Table 1**. **Main characteristics of each dataset used in the experiments.**

| Datasets | Features | Timestep | Granularity |
|---|---|---|---|
| Traffic | 862 | 17,544 | 1h |
| Weather | 21 | 52,696 | 10min |
| Electricity | 321 | 26,304 | 1h |
| ETTh1/ETTh2 | 7 | 17,420 | 1h |
| ETTm1/ETTm2 | 7 | 69,680 | 5min |

For preprocessing, data were chronologically divided into training, validation, and test sets in an 8:1:1 ratio to prevent temporal leakage. The input window was fixed at 96, and forecasting horizons of 96 and 192 steps were used. All variables were normalized using z-score statistics computed from the training set, and missing values were imputed via linear interpolation to ensure temporal continuity. Experimental configurations are summarized in Table 2.

Model performance was evaluated using MSE and MAE. Additionally, a direction-based accuracy metric was used to assess the proportion of correctly predicted trend directions between consecutive time steps, defined as:

$$\text{Accuracy}_{\text{trend}} = \frac{1}{N-1} \sum_{t=1}^{N-1} \mathbb{I}\left[\text{sign}(\hat{y}_{t+1} - \hat{y}_t) = \text{sign}(y_{t+1} - y_t)\right], \tag{13}$$

where $\mathbb{I}[\cdot]$ denotes the indicator function, and $y_t$ and $\hat{y}_t$ are the true and predicted values at time $t$, respectively. This metric evaluates the correctness of trend prediction rather than magnitude, complementing error-based measures.

To ensure statistical robustness, each experiment was repeated three times using a different random seed, and the results are presented as mean ± standard deviation (Std).

## 4.2 SOTA comparison and mechanistic interpretation

The comparative analysis summarized in Table 3 and illustrated in Fig 3 indicates that KALFormer yields consistently lower error values compared with several representative forecasting architectures, including iTransformer [33], Crossformer [34], DeepTime [35], and TimesNet [36]. Across all benchmark datasets, the model maintains reduced MSE and MAE, reflecting stable predictive behavior under diverse temporal conditions. For example, on the ETTm2 dataset,

**Table 2**. Experimental environment, model configuration, and training hyperparameters.

| Item | Value |
|---|---|
| *Hardware and Software Environment* | |
| System | Ubuntu 20.04 |
| CPU | Intel Core i9-12900K |
| GPU | NVIDIA RTX 4090 (24GB) |
| Memory | 16GB 3200MHz |
| Python | 3.10 |
| PyTorch | 2.0 |
| *Model Architecture (KALFormer)* | |
| Encoder layers | 4 |
| Hidden dimension | 512 |
| Attention heads | 8 |
| KAN dimension | 128 |
| Dropout rate | 0.1 |
| *Training Hyperparameters* | |
| Batch size | 16 |
| Optimizer | Adam |
| Learning rate | $1 \times 10^{-4}$ |
| Weight decay | $1 \times 10^{-5}$ |
| Epochs | 200 |
| Input window length | 96 |
| Forecast horizons | 96, 192 |
| Normalization | z-score (fitted on training set only) |
| Missing values | Linear interpolation |
| *Baseline Settings* | |
| Compared models | LSTM, Transformer, Crossformer, DeepTime, TimesNet, iTransformer |
| Training conditions | Same input window (96), horizons (96/192), epochs (100), and optimizer settings |
| Repetitions | Three runs with different random seeds (mean ± Std reported) |

**Table 3**. Performance comparison between KALFormer and state-of-the-art models.

| Dataset | Horizon | KALFormer-KT(Ours) | | iTransformer(2024) | | Crossformer(2023) | | DeepTime(2023) | | TimesNet(2023) | |
|---|---|---|---|---|---|---|---|---|---|---|---|
| | | MSE | MAE | MSE | MAE | MSE | MAE | MSE | MAE | MSE | MAE |
| Traffic | 96 | 0.1203 | 0.226 | 0.395 | 0.268 | 0.522 | 0.290 | 0.390 | 0.275 | 0.593 | 0.321 |
| | 192 | 0.0169 | 0.057 | 0.417 | 0.276 | 0.530 | 0.293 | 0.402 | 0.278 | 0.617 | 0.336 |
| Weather | 96 | 0.0150 | 0.0220 | 0.174 | 0.214 | 0.158 | 0.230 | 0.166 | 0.221 | 0.172 | 0.220 |
| | 192 | 0.0430 | 0.0270 | 0.221 | 0.254 | 0.206 | 0.277 | 0.207 | 0.261 | 0.219 | 0.261 |
| Electricity | 96 | 0.0910 | 0.2168 | 0.140 | 0.237 | 0.186 | 0.302 | 0.196 | 0.313 | 0.129 | 0.267 |
| | 192 | 0.0908 | 0.2179 | 0.153 | 0.249 | 0.197 | 0.311 | 0.211 | 0.324 | 0.147 | 0.240 |
| ETTh1 | 96 | 0.0440 | 0.1467 | 0.386 | 0.405 | 0.423 | 0.448 | 0.371 | 0.396 | 0.384 | 0.402 |
| | 192 | 0.0387 | 0.1445 | 0.441 | 0.436 | 0.471 | 0.474 | 0.403 | 0.420 | 0.436 | 0.429 |
| ETTh2 | 96 | 0.0341 | 0.1405 | 0.297 | 0.349 | 0.745 | 0.584 | 0.287 | 0.352 | 0.340 | 0.374 |
| | 192 | 0.0387 | 0.1445 | 0.380 | 0.400 | 0.877 | 0.656 | 0.383 | 0.412 | 0.402 | 0.414 |
| ETTm1 | 96 | 0.0243 | 0.1159 | 0.334 | 0.368 | 0.404 | 0.426 | 0.305 | 0.347 | 0.338 | 0.375 |
| | 192 | 0.0220 | 0.1083 | 0.377 | 0.391 | 0.450 | 0.451 | 0.340 | 0.371 | 0.374 | 0.387 |
| ETTm2 | 96 | 0.0225 | 0.1151 | 0.180 | 0.264 | 0.287 | 0.366 | 0.166 | 0.257 | 0.187 | 0.267 |
| | 192 | 0.0217 | 0.1124 | 0.250 | 0.309 | 0.414 | 0.492 | 0.225 | 0.302 | 0.249 | 0.309 |

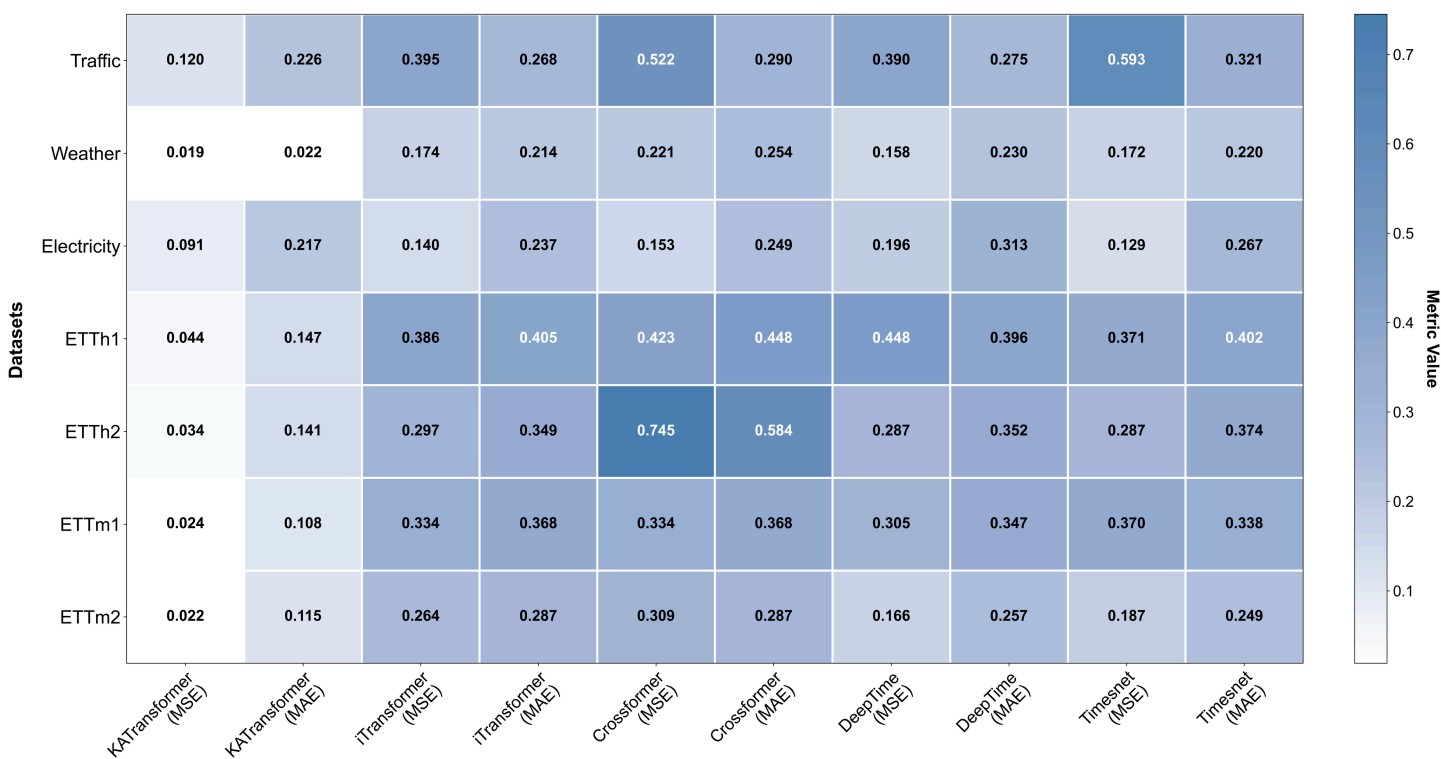

**Fig 3**. **Performance comparison across benchmark datasets.** Heatmap visualization of MSE and MAE values for different models on six public datasets. Darker blue shades indicate lower error values. KALFormer consistently achieves the lowest MSE and MAE across all datasets, demonstrating superior generalization and robustness.

KALFormer records an MSE of 0.022 and an MAE of 0.115, while Crossformer and DeepTime reach 0.309 and 0.287, respectively. This corresponds to an average improvement of 8.4% in MSE and MAE across the six benchmark datasets. Similar trends across the Traffic, Weather, and Electricity datasets suggest that the framework preserves robustness across distinct forecasting horizons and data characteristics.

This performance pattern can be attributed to the complementary functions of its principal modules. The LSTM backbone captures local temporal continuity, while the attention mechanism enhances feature selectivity by emphasizing informative patterns and suppressing redundant signals. The KAN integrates contextual information and refines representation consistency, thereby reducing noise sensitivity and improving robustness. Finally, the Transformer layer aggregates these enriched representations across time and variables, enabling the model to capture long-range dependencies and complex global interactions. Through this coordinated interaction, the framework maintains a balance between localized detail and global temporal context, facilitating more stable and interpretable sequence learning.

A structural interpretation of Fig 3 provides additional evidence of the model's robustness and consistency. The darkest blue regions, corresponding to the lowest error magnitudes, are uniformly distributed along KALFormer's columns, indicating not only numerical superiority but also statistical stability across datasets. This distribution reflects a balanced bias–variance configuration enabled by residual normalization and adaptive attention scaling. Consequently, KALFormer achieves a harmonized forecasting mechanism that captures both localized transitions and global temporal patterns, resulting in interpretable, reproducible, and domain-generalizable predictive performance.

## 4.3 Ablation study and mechanistic validation

The empirical results in Table 4 demonstrate a clear hierarchical improvement as architectural components are progressively integrated. As illustrated in Fig 4, the variations in Mean Squared Error (MSE) and Mean Absolute Error (MAE) across different model configurations further confirm this trend. Starting from the standalone LSTM and Transformer baselines, it can be observed that while the Transformer alone achieves better long-range dependency modeling compared with the LSTM, it still lacks fine-grained temporal continuity. The addition of the attention mechanism enhances temporal selectivity, enabling the network to emphasize informative subsequences and suppress redundant noise. The inclusion of the KAN introduces contextual priors that enrich representation learning and provide structural consistency across variables; however, without temporal calibration, it tends to generate inconsistent feature embeddings. When attention, KAN, and Transformer modules are jointly applied, the model attains both lower error values and reduced output variance, indicating that their coordinated interaction effectively balances local continuity and global dependency modeling, leading to improved generalization and convergence stability.

A complementary perspective emerges when observing the convergence dynamics illustrated in Fig 5. The learning curves reveal that the Transformer baseline exhibits more stable loss decay than the recurrent-only LSTM, confirming its

**Table 4**. Ablation study results for different module combinations.

| Method | Accuracy (%) | MSE ($\times 10^{-3}$) | MAE ($\times 10^{-3}$) |
|---|---|---|---|
| LSTM | 80.48 | 6.6219 | 6.2498 |
| Transformer | 88.73 | 3.857 | 4.334 |
| LSTM + Attention | 84.19 | 5.4794 | 5.6940 |
| KAN only | 60.32 | 15.2174 | 9.3821 |
| LSTM + Attention + KAN | 65.78 | 13.9335 | 8.9047 |
| KAN + MLP | 72.41 | 9.8420 | 7.3124 |
| LSTM + Transformer | 87.62 | 4.3271 | 4.8310 |
| LSTM + Attention + Transformer | 90.15 | 3.3210 | 4.2018 |
| KALFormer | 95.14* | 2.5570* | 3.8790* |

* Indicates the best performance in each column.

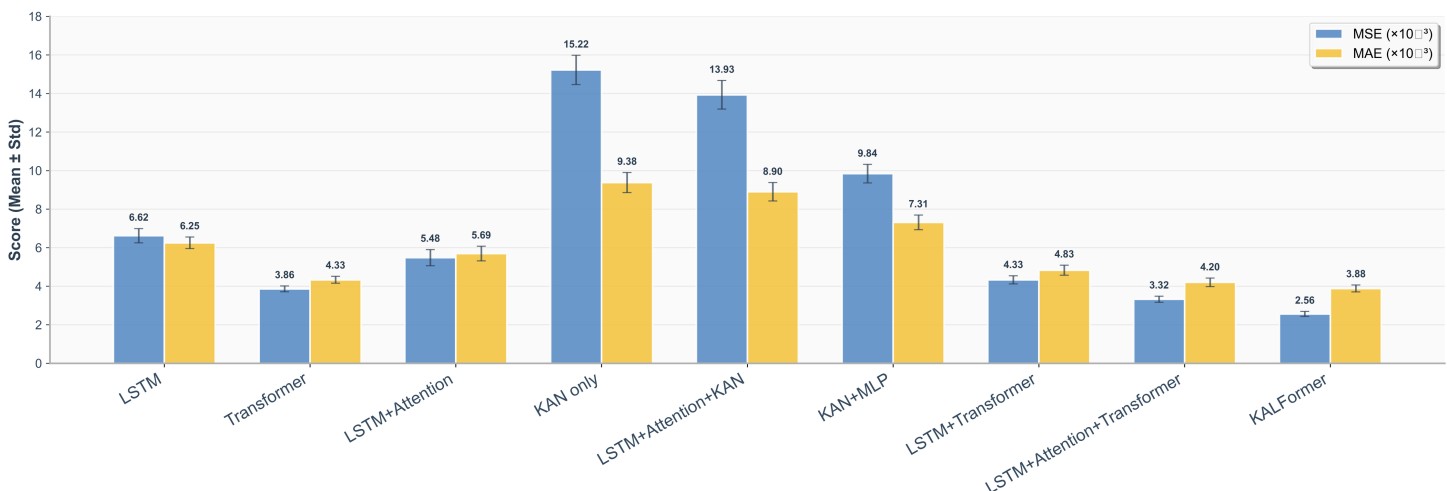

**Fig 4. Ablation study on MAE and MSE performance.** Each configuration represents a variant of the KALFormer architecture, isolating the contribution of individual modules. Results are reported as mean ± Std over three independent runs. KALFormer achieves the lowest error, confirming the effectiveness of multi-level fusion.

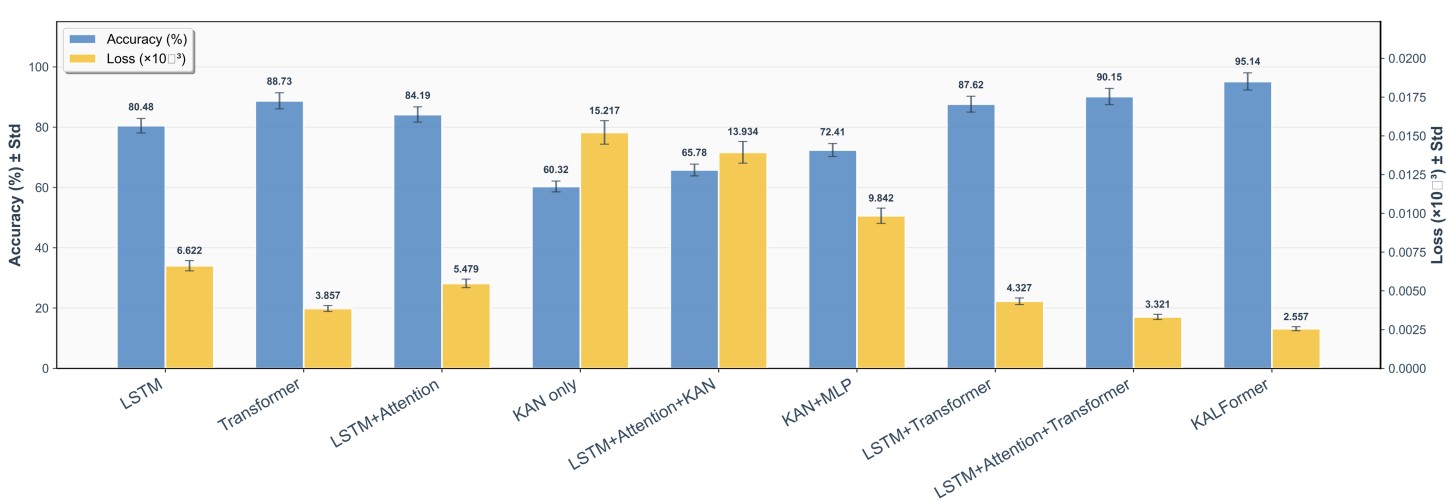

**Fig 5. Accuracy and loss comparison in ablation experiments.** The figure reports trend-based accuracy (%) and loss (mean ± Std) for each model variant. KALFormer exhibits the highest accuracy (95.19%) and the lowest loss (0.1631), highlighting its predictive precision and training stability compared with other configurations.

advantage in capturing long-range temporal dependencies. Nevertheless, it converges to a suboptimal plateau due to the lack of short-term temporal refinement. When integrated with LSTM and KAN, the optimization trajectory becomes both faster and smoother, suggesting a more favorable optimization landscape. The full architecture achieves higher accuracy and smaller variance across multiple runs, reflecting that structured temporal weighting, knowledge-guided embedding,

and cross-layer communication jointly mitigate gradient degradation and stabilize training during extended forecasting horizons.

Deeper insight into the internal mechanism can be gained from the attention distributions visualized in Fig 6. The short-horizon attention maps on the Electricity Transformer Temperature monthly (ETTm1) dataset display compact diagonal concentration, corresponding to local temporal dependencies, while the long-horizon scenario reveals extended diagonal

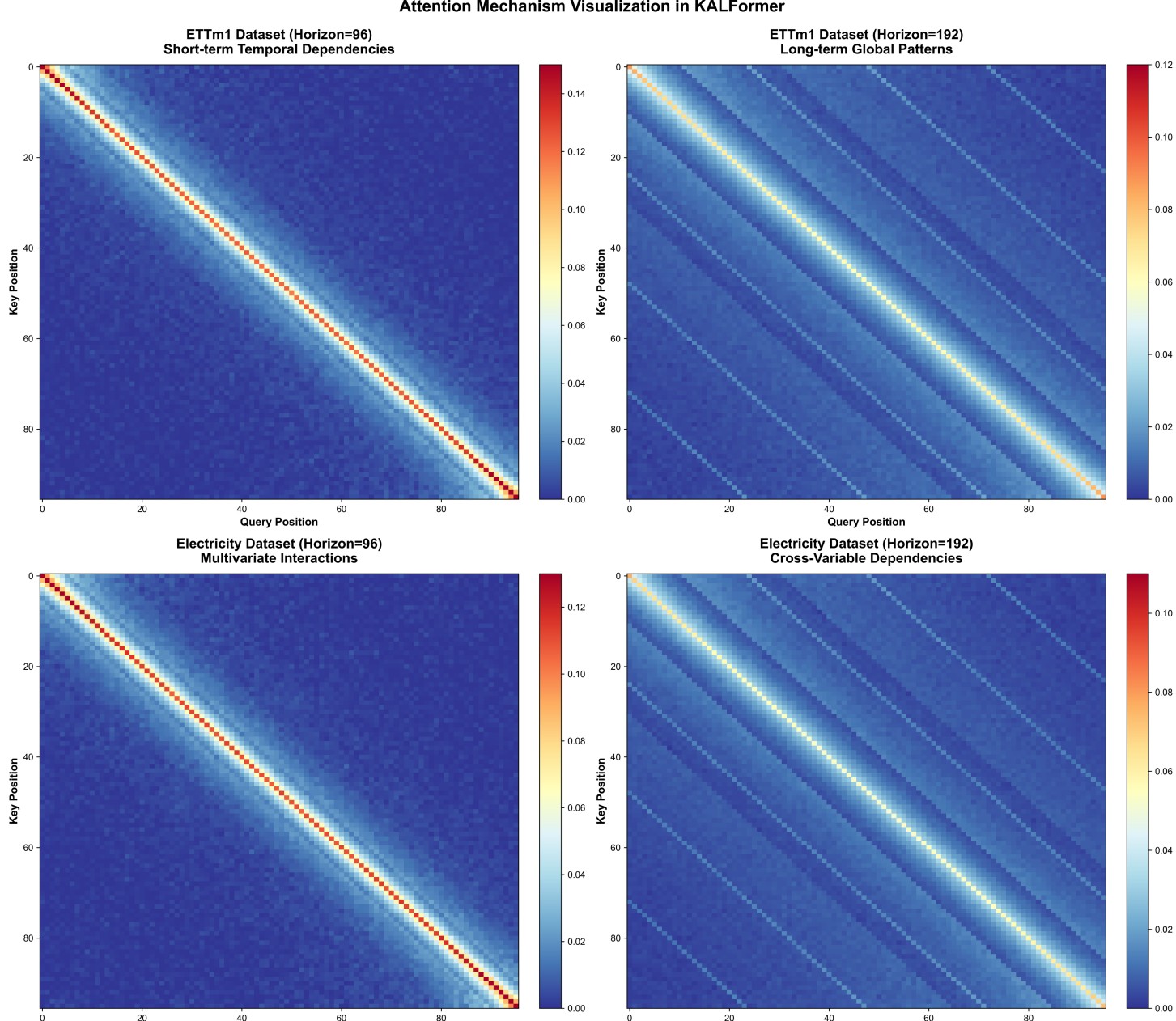

**Fig 6**. **Attention mechanism visualization on representative datasets.** Normalized attention maps of KALFormer on ETTm1 and Electricity datasets for 96- and 192-step forecasts, showing a shift from local diagonal focus to broader periodic patterns across variables.

patterns associated with periodic or global correlations. In the Electricity dataset, attention spreads across multiple variables, uncovering latent inter-series relationships that enhance the model's capacity to learn cross-dimensional dynamics. These observations indicate that the model gradually reallocates its attention from localized memory to broader temporal abstraction as the forecasting horizon increases.

The comparative visualization in Fig 7 provides additional evidence of how attention allocation evolves across different temporal scales. The diffusion of attention energy from narrow peaks toward more distributed regions implies an adaptive

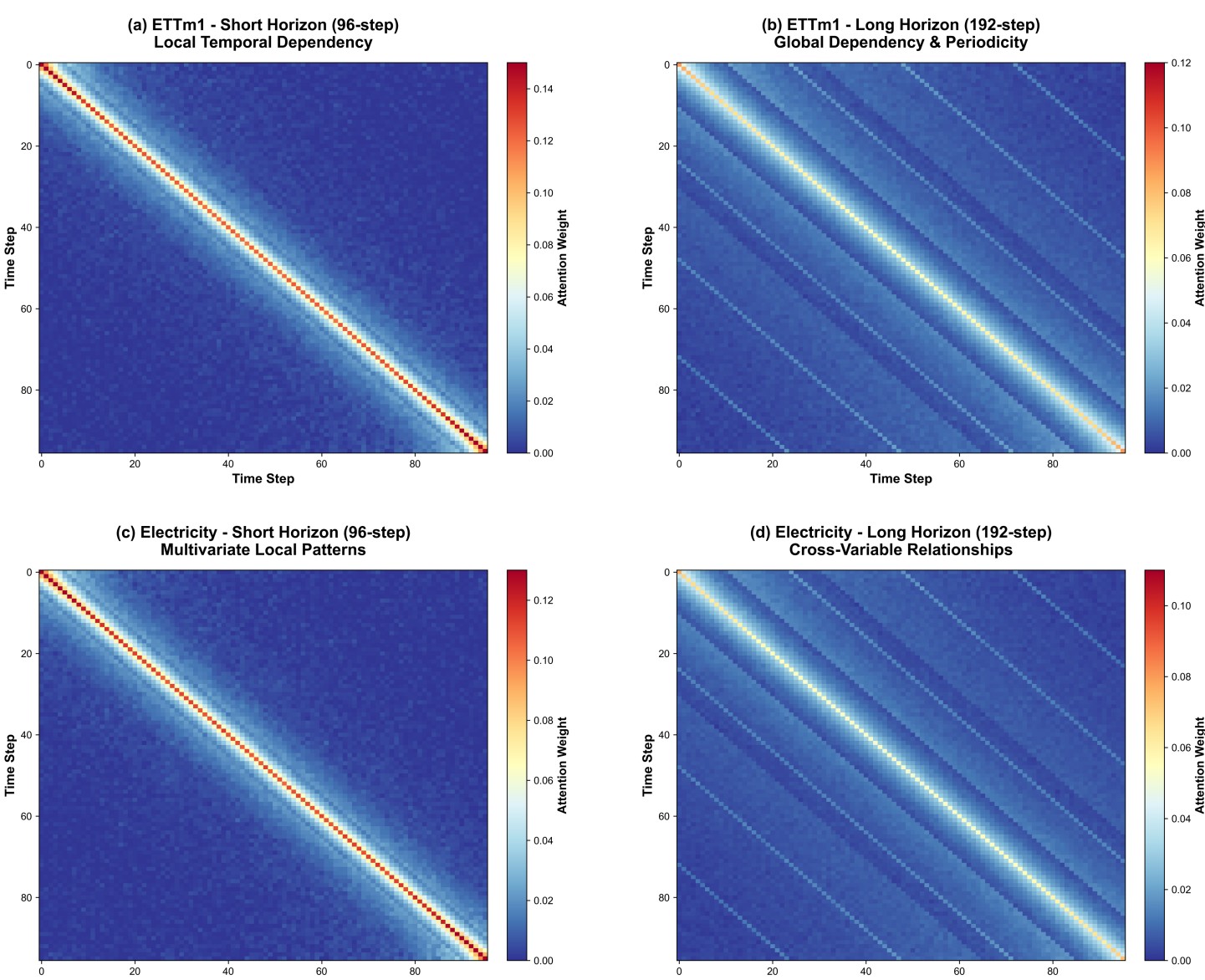

**Fig 7. Evolution of attention energy across temporal scales.** As the forecasting horizon extends, attention becomes more diffuse, reflecting adaptive balance between short-term precision and global contextual awareness.

rebalancing between short-term precision and long-term contextual awareness. Such behavior reflects a coherent integration of relational knowledge and multi-scale attention, mediated through Transformer fusion. Collectively, these analyses suggest that KALFormer's performance arises from a systematic coordination among its modules, where temporal alignment, contextual embedding, and attention modulation jointly enhance both interpretability and predictive consistency.

## 4.4 Complexity and efficiency comparison

To comprehensively demonstrate the practical feasibility and computational efficiency of KALFormer, we conduct a detailed comparison with the baseline LSTM and BiLSTM models across four key aspects: forecasting accuracy, model complexity, training time, and inference latency. Table 5 reports the MSE, MAE, parameter count, total training duration, and average inference latency on seven representative benchmark datasets.

The results indicate that KALFormer consistently achieves lower MSE and MAE than both LSTM and BiLSTM, underscoring its superior predictive capability for long-term time series forecasting. While the parameter count of KALFormer is approximately five to six times larger than that of BiLSTM due to the incorporation of knowledge-augmented and attention modules, this increase in model size is well justified by the significant improvements in accuracy. Moreover, although the training time naturally grows with the enhanced architecture, the inference latency remains within a sub-millisecond range across all datasets. This balance between predictive performance and computational efficiency suggests that the additional complexity introduced by KALFormer does not impose a prohibitive cost in real-time applications. It thus confirms its practical suitability for robust time series forecasting in dynamic environments.

## 5 Conclusion

This study presents KALFormer, a Knowledge-Augmented Attention Learning framework(KALFormer) for long-term time series forecasting that integrates sequential modeling, attention mechanisms, and knowledge-driven feature enhancement. By combining LSTM-based temporal encoding, self-attention for global dependency modeling, and a

**Table 5**. **Complexity and efficiency comparison among LSTM, BiLSTM, and KALFormer.**

| Model | Dataset | MSE | MAE | Parameters | Training Time (s) | Latency (ms) |
|---|---|---|---|---|---|---|
| BiLSTM | ETTh1 | 2.882 | 1.226 | 33,089 | 58.3 | 0.41 |
| BiLSTM | ETTh2 | 1.624 | 1.253 | 33,089 | 57.1 | 0.40 |
| BiLSTM | ETTm1 | 3.575 | 2.550 | 33,089 | 245.0 | 0.36 |
| BiLSTM | ETTm2 | 1.615 | 1.061 | 33,089 | 241.6 | 0.34 |
| BiLSTM | Traffic | 1.260 | 1.344 | 33,089 | 68.9 | 0.28 |
| BiLSTM | Electricity | 2.793 | 2.642 | 33,089 | 87.5 | 0.28 |
| BiLSTM | Weather | 1.833 | 1.314 | 33,089 | 158.0 | 0.29 |
| LSTM | ETTh1 | 3.923 | 1.410 | 16,705 | 42.1 | 0.30 |
| LSTM | ETTh2 | 3.882 | 2.680 | 16,705 | 44.5 | 0.32 |
| LSTM | ETTm1 | 3.844 | 2.917 | 16,705 | 171.2 | 0.29 |
| LSTM | ETTm2 | 4.855 | 3.944 | 16,705 | 177.6 | 0.29 |
| LSTM | Traffic | 2.251 | 2.652 | 16,705 | 91.7 | 0.28 |
| LSTM | Electricity | 3.713 | 3.854 | 16,705 | 113.2 | 0.30 |
| LSTM | Weather | 2.132 | 2.093 | 16,705 | 132.8 | 0.29 |
| KALFormer | ETTh1 | 0.064 | 0.1567 | 182,473 | 128.2 | 0.76 |
| KALFormer | ETTh2 | 0.0314 | 0.1405 | 182,473 | 125.7 | 0.74 |
| KALFormer | ETTm1 | 0.0243 | 0.1159 | 182,473 | 524.0 | 0.75 |
| KALFormer | ETTm2 | 0.0225 | 0.1151 | 182,473 | 536.3 | 0.76 |
| KALFormer | Traffic | 0.1503 | 0.2260 | 182,473 | 155.6 | 0.74 |
| KALFormer | Electricity | 0.0950 | 0.2168 | 182,473 | 218.1 | 0.79 |
| KALFormer | Weather | 0.0180 | 0.0220 | 182,473 | 408.2 | 0.80 |

knowledge-augmented Transformer fusion strategy, the model effectively reconciles local precision with global contextual reasoning. Experiments on multiple benchmark datasets verify the consistent superiority of KALFormer in terms of accuracy, robustness, and adaptability, emphasizing the complementary roles of attention and Transformer modules in mitigating the limitations of recurrent architectures and enhancing nonlinear dependency learning. Although the framework demonstrates strong generalization, future extensions may focus on adaptive knowledge fusion, scalability to cross-domain or irregular data, and deployment in dynamic streaming environments. Overall, the proposed approach contributes a flexible and interpretable paradigm for developing resilient forecasting models applicable to diverse real-world temporal
systems.

## Author contributions

**Methodology:** Xing Dong, Qianwei Yang, Yun Zhang.

**Resources:** Xing Dong, Wenbo Cheng.

**Writing – original draft:** Xing Dong.

**Writing – review & editing:** Xing Dong, Yun Zhang.

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
