## [Decision Letter · Decision Letter 0]

6 Sep 2025

PONE-D-25-38730KALFormer: Knowledge-Augmented Attention Learning for Long-Term Time Series Forecasting with TransformerPLOS ONE

Dear Dr. Zhang,

Thank you for submitting your manuscript to PLOS ONE. After careful consideration, we feel that it has merit but does not fully meet PLOS ONE’s publication criteria as it currently stands. Therefore, we invite you to submit a revised version of the manuscript that addresses the points raised during the review process. Both reviewers raised important points: I strongly recommend following the suggestions of Reviewer #1, particularly the ones about technical issues (equations, methodology, analysis of results), but also consider answering the more general comments of Reviewer #2 or providing counterarguments where appliable. Also please review your paper for small typos and mistakes, e.g: addresses of authors sometimes have spaces before/after the ZIP code, sometimes not. Some references' titles contains abbreviations and acronyms: should those be capitalized? E.g. reference 21 (A3t-gcn), reference 12 (Multi-head attention). Also, "Recurrent Neural Network" is capitalized in reference 25 but not in others, e.g. 30. Just to be safe, please consider using a reference management software.  Please also see the reviewers' comments on the quality of the figures, For example, figure 3 is blurry and the numbers on the table are tiny.

We look forward to receiving your revised manuscript.

Kind regards,

Rafael Duarte Coelho dos Santos, Ph.D.

Academic Editor

PLOS ONE

Journal Requirements:

National Natural Science Foundation of China GuizhouUniversity of Traditional Chinese MMedicine

4. Thank you for uploading your study's underlying data set. Unfortunately, the repository you have noted in your Data Availability statement does not qualify as an acceptable data repository according to PLOS's standards.

5. We note you have included a table to which you do not refer in the text of your manuscript. Please ensure that you refer to Table 3 in your text; if accepted, production will need this reference to link the reader to the Table.

Additional Editor Comments:

Reviewer #1:

Please verify the detailed suggestions on technical aspects of the paper, particularly formulation and statistical analysis, and also make sure the public code is the one related to the paper and fully documented.

Reviewer #2:

Please check and address the comments, in particular comment 1 (related to the whole approach in general -- may be necessary to compare with the current state of the art in time series forecasting) and comment 5 (more details on the interpretation pf the results).

Reviewers' comments:

Reviewer's Responses to Questions

**Comments to the Author**

1. Is the manuscript technically sound, and do the data support the conclusions?

Reviewer #1: Partly

Reviewer #2: Partly

2. Has the statistical analysis been performed appropriately and rigorously?

Reviewer #1: No

Reviewer #2: No

3. Have the authors made all data underlying the findings in their manuscript fully available?

Reviewer #1: Yes

Reviewer #2: Yes

4. Is the manuscript presented in an intelligible fashion and written in standard English?

Reviewer #1: Yes

Reviewer #2: No

5. Review Comments to the Author

Reviewer #1: The manuscript presents KALFormer, a hybrid model integrating LSTM, multi-head attention, knowledge-augmented networks (KAN), and Transformer modules for long-term time series forecasting. The idea of combining sequential modeling with external knowledge augmentation is novel and potentially valuable. However, in its current form the paper has several important issues that limit reproducibility, clarity, and rigor. Below I outline my detailed comments.

1. Technical soundness

The overall framework is reasonable and produces encouraging results across multiple benchmark datasets (Traffic, Electricity, Weather, ETTh1/2, ETTm1/2).

However, there are issues with the mathematical formulation. Equation (3) misrepresents the standard scaled dot-product attention. The correct formulation is:

Attention(,,)=softmax(⊤)Attention(Q,K,V)=softmax(d QK⊤ )V

The manuscript currently places the softmax outside the multiplication with V, which is not correct. This should be revised to avoid confusion and ensure technical accuracy.

The KAN component is under-specified. It is not clear how the knowledge graph is constructed for each dataset:

What constitutes nodes and edges?

How are edge weights computed (correlation, distance, domain rules)?

Are graphs dataset-specific or universal?

How many GCN layers are used and how is the fusion weight initialized and trained?

Without these details, the contribution of the “knowledge” component cannot be independently validated or reproduced.

2. Statistical analysis

The manuscript reports MSE and MAE, which are standard metrics for regression tasks. However, there are inconsistencies between results across different tables. For example, the ETTh1 MSE in Table 3 differs significantly from the value reported in Table 5. Please clarify whether these correspond to different prediction horizons, different data splits, or a reporting error.

In Table 4 (ablation study), an “Accuracy (%)” metric is introduced. Accuracy is not standard for time series regression, and no definition is provided. Does this refer to directional accuracy (predicting the correct trend) or tolerance-based accuracy (predictions within a margin)? This must be defined formally in the Methods section. If it cannot be rigorously justified, it should be replaced with standard error metrics (RMSE, MAPE, or directional metrics explicitly stated).

There is no discussion of statistical significance or confidence intervals. To strengthen the claims of superiority over baselines, the authors should consider including repeated runs with error bars or tests such as the Diebold–Mariano test.

3. Data and code availability

The code is shared via GitHub, which is commendable. However, the manuscript describes KALFormer but the linked repository is titled FusionLSTM_KT. It is unclear whether this repository contains the exact version of KALFormer used to generate the reported results. To satisfy reproducibility requirements, please ensure the repository contains:

A clearly named KALFormer implementation.

Configuration files and scripts for each dataset/horizon.

Instructions to reproduce Tables 3–5 exactly.

The Data Availability Statement does not meet PLOS ONE policy. The Kaggle link contains a misspelled username (“anonuymous”) and has no persistent DOI. PLOS requires that all underlying data be deposited in a stable, publicly accessible repository (e.g., Zenodo, OSF, Figshare, or Kaggle with DOI). Preprocessing scripts should also be provided so other researchers can replicate the train/validation/test splits and normalization steps.

4. Experimental setup and reproducibility

The manuscript notes that datasets are split 8:1:1, but does not specify if the split is time-ordered (which is typical for forecasting tasks) or random. This affects the validity of the evaluation.

Details such as input sequence length (look-back window), output horizons (96, 192, etc.), and normalization (per-feature vs per-series, fit on training only or across the whole dataset) should be reported.

Missing value handling is not discussed. Were missing data imputed, and if so, how?

Hyperparameters such as learning rate, dropout, number of epochs, batch size, optimizer, and weight decay should be provided. Currently, only the hardware setup is described.

Providing a summary table of these details would substantially improve reproducibility.

5. Ablation study and interpretation

The ablation study shows that KAN + Attention (without Transformer) reduces performance, but the full model improves once the Transformer is added. This is attributed to “feature inconsistency,” but this explanation is speculative without additional baselines.

A stronger comparison would include:

A “KAN only” baseline.

A “KAN + simple concatenation + MLP” baseline.

This would clarify whether the Transformer specifically enables effective integration of KAN features.

The narrative should also reconcile why KAN appears to degrade performance in some ablations but still improves the full model.

6. Presentation and writing

The manuscript is generally intelligible and written in clear English. However, minor issues remain:

Typos such as “anonuymous” and “MMedicine.”

Inconsistent capitalization of terms (KAN, Transformer, MHA).

Figures should be provided as high-resolution images per journal guidelines.

Correcting these will improve readability.

7. Funding and policy compliance

The funding statement currently lists organizations but does not provide all details required by PLOS ONE (initials of funded authors, grant numbers, funder URLs, and explicit statement of funders’ role). Please expand this section accordingly.

Summary:

This manuscript proposes a creative hybrid model for long-term time series forecasting and shows promising empirical results. However, major revisions are required before the manuscript can be considered for publication. The main issues involve data and code availability, mathematical correctness, statistical rigor, and insufficient methodological detail for reproducibility. Addressing these points will significantly strengthen the work and ensure compliance with PLOS ONE standards.

Reviewer #2: The authors have focused on time series forecasting, an area that has already seen extensive research and progress.

The reviewer has the following concerns:

1. The problem described in the paper is inherently a multivariate time series forecasting task, for which significant research already exists. However, starting from the introduction, the authors frame the problem as forecasting a target series while treating other time series merely as influencing factors. This approach raises questions regarding the fairness of the problem analysis and validation.

2. In the validation of the models, both the proposed model and the baseline comparisons lack details regarding model architecture parameters and key configuration settings. As a result, the fairness of the results cannot be guaranteed.

3. The use of LSTM for feature extraction from the time series before applying the Transformer may seem reasonable. However, the reviewers question whether this modification is truly beneficial—given that the Transformer itself is capable of extracting temporal features and its strengths lie in self-attention and cross-attention mechanisms. From the perspectives of performance and effectiveness, is such a design choice justified?

4. The connections between modules in the proposed model need further refinement, particularly in terms of the input and output features.

5. The analysis and discussion of the experimental findings are overly brief, especially in explaining the reasons behind the performance improvement. More in-depth interpretation is needed.

6. PLOS authors have the option to publish the peer review history of their article (what does this mean?). If published, this will include your full peer review and any attached files.

Reviewer #1: No

Reviewer #2: No

---

## [Author Response · Author response to Decision Letter 1]

6 Nov 2025

Dear Editor and Reviewers,

We sincerely thank you for your valuable feedback and constructive suggestions. All comments from both reviewers and the editorial office have been carefully addressed in the revised manuscript. A detailed, point-by-point response outlining all modifications has been provided in the attached file “Response to Reviewers.docx.”

We truly appreciate your time and effort in reviewing our work and believe that the revisions have significantly improved the quality and clarity of the manuscript.

---

## [Editor Report · Decision Letter 1]

17 Nov 2025

KALFormer: Knowledge-Augmented Attention Learning for Long-Term Time Series Forecasting with Transformer

PONE-D-25-38730R1

Dear Dr. Zhang,

We’re pleased to inform you that your manuscript has been judged scientifically suitable for publication and will be formally accepted for publication once it meets all outstanding technical requirements.

Kind regards,

Rafael Duarte Coelho dos Santos, Ph.D.

Academic Editor

PLOS ONE

Additional Editor Comments (optional):

The corrections and improvements are satisfactory.
---

## [Editor Report · Acceptance letter]

PONE-D-25-38730R1

PLOS One

Dear Dr. Zhang,

I'm pleased to inform you that your manuscript has been deemed suitable for publication in PLOS One. Congratulations! Your manuscript is now being handed over to our production team.

Kind regards,

on behalf of

Dr. Rafael Duarte Coelho dos Santos

Academic Editor

PLOS One